# Pricing Product Options and Using Them to Complete Markets for Functions of Two Underlying Asset Prices [†]

**Dilip B. Madan** [1,*] and **King Wang** [2]

1　Robert H. Smith School of Business, University of Maryland, College Park, MD 20742, USA
2　Derivative Product Strats, Morgan Stanley, 1585 Broadway, 5th Floor, New York, NY 10036, USA; king.wang@morganstanley.com
*　Correspondence: dbm@umd.edu
†　This paper is the private opinion of the authors and does not necessarily reflect the policy and views of Morgan Stanley.

**Abstract:** Options paying the product of put and/or call option payouts at different strikes on two underlying assets are observed to synthesize joint densities and replicate differentiable functions of two underlying asset prices. The pricing of such options is undertaken from three perspectives. The first perspective uses a geometric two-dimensional Brownian motion model. The second inverts two-dimensional characteristic functions. The third uses a bootstrapped physical measure to propose a risk charge minimizing hedge using options on the two underlying assets. The options are priced at the cost of the hedge plus the risk charge.

**Keywords:** multivariate bilateral gamma; fast Fourier transform; distorted expectations; acceptable risks

## 1. Introduction

Option markets considerably enhance the collection of functions of the stock price at the option maturity that may be purchased or sold. In the absence of options, the only functions one can access are straight lines with bonds delivering the intercept and stocks the slope. Options allow one to change the slope and access any twice differentiable function of the stock price as shown, for example, in Carr and Madan (1998).

Functions that have been priced this way include the logarithm of the S&P 500 index a month later in arriving at the *VIX* index. The *CBOE* skew prices the first three powers of the logarithm to build the Skew index. Others have used the procedure to study the risk neutral kurtosis and its effects (Bakshi et al. (2003)). It is also known from Breeden and Litzenberger (1978) that option prices synthesize risk neutral densities and permit the recovery of the density from the second derivative of option prices.

These results have natural extensions to two dimensions when one considers functions of a pair of stock prices at maturity. Functions differentiable four times may be replicated provided one trades product options. The payoff on a product option is the product of payoffs on two standard options be they puts or calls. Furthermore, joint densities may be synthesized from the prices of such product options. Consider, for example, a product call with strikes $K_1, K_2$. For stock prices $S_1, S_2$ at maturity, the payoff or value at maturity $T$ is

$$C(S_1, S_2, T; K_1, K_2) = (S_1 - K_1)^+ (S_2 - K_2)^+. \tag{1}$$

For a risk neutral density $f(S_1, S_2)$, the price of the product call $w(K_1, K_2)$ for an interest rate $r$ is given by

$$w(K_1, K_2) = e^{-rT} \int_{K_1}^{\infty} \int_{K_2}^{\infty} (S_1 - K_1)(S_2 - K_2)f(S_1, S_2)dS_1 dS_2.$$

Differentiation shows that

$$\frac{\partial^4}{\partial K_1^2 \partial K_2^2} e^{rT} w(K_1, K_2) = f(K_1, K_2). \tag{2}$$

Other approaches to such recovery using basket option prices include Lipton (2001) and Carr and Laurence (2011).

Product options, therefore, play a critical role when one considers functions of pairs of prices. We, therefore, consider the pricing of such options. The first context is a simple generalization of the Black and Scholes (1973) and Madan (1973) geometric Brownian motion model to a bivariate normal density for the logarithm of the two stock prices. The product options are then priced using bivariate normal distribution functions and an assumed correlation value. The bivariate normal has normal marginals that do not fit the marginal risk neutral densities.

However, there is a large class of pure jump Lévy process models that fit option prices at each maturity. Among them is the three parameter variance gamma model of Madan and Seneta (1990), Madan et al. (1998). Recently, this model has been generalized to the four parameter bilateral gamma model of Küchler and Tappe (2008) that also fits marginal risk neutral densities. The bilateral gamma model is used to fit the two risk neutral marginals. Recent work reported in Madan and Schoutens (2020), Madan (2020) and Madan and Wang (2020a) formed multivariate densities consistent with prespecified bilateral gamma models with two additional dependency parameters. The multivariate bilateral gamma model has a closed form characteristic function.

The methods of Carr and Madan (1999) are extended to employ the two dimensional fast Fourier transform to price product options. In addition, methods reported in Madan and Wang (2020b) may be employed to price product options from the physical measures using the options in the two markets as hedge instruments.

The outline of the rest of the paper is as follows. Section 2 develops the two dimensional replication result. Section 3 presents the bivariate log normal pricing of product options. The two dimensional fast Fourier inversion of suitably modified product options is taken up in Section 4. The multivariate bilateral gamma model is described in Section 5. Sample computations and the effects on product option prices of changes in the dependency parameters of the multivariate bilateral gamma model are illustrated in Section 6. Physical measure pricing of product options is formulated and implemented in Section 7. Finally, Section 8 concludes.

## 2. Product Options and Functional Replication

Let $g(x, y)$ be a sufficiently smooth function of two variables. It may then be written as the integral of its second order cross partials as follows. For an arbitrary reference point $(x_0, y_0)$

$$g(x, y) = g(x_0, y) + g(x, y_0) - g(x_0, y_0) + \int_{x_0}^{x} \int_{y_0}^{y} \frac{\partial^2 g(u, v)}{\partial x \partial y} du dv. \tag{3}$$

Now, we apply this result to the second cross partial itself to write

$$
\begin{aligned}
g(x,y) &= g(x_0,y) + g(x,y_0) - g(x_0,y_0) \\
&\quad + \int_{x_0}^{x} \int_{y_0}^{y} \left[ \begin{array}{c} \frac{\partial^2 g(x_0,v)}{\partial x \partial y} + \frac{\partial^2 g(u,y_0)}{\partial x \partial y} - \frac{\partial^2 g(x_0,y_0)}{\partial x \partial y} \\ + \int_{x_0}^{u} \int_{y_0}^{v} \frac{\partial^4 g(a,b)}{\partial x^2 \partial y^2} da\, db \end{array} \right] du\, dv \qquad (4) \\
&= g(x_0,y) + g(x,y_0) - g(x_0,y_0) \\
&\quad + (x - x_0) \left[ \frac{\partial g(x_0,y)}{\partial x} - \frac{\partial g(x_0,y_0)}{\partial x} \right] + (y - y_0) \left[ \frac{\partial g(x,y_0)}{\partial y} - \frac{\partial g(x_0,y_0)}{\partial y} \right] \\
&\quad - \frac{\partial^2 g(x_0,y_0)}{\partial x \partial y} (x - x_0)(y - y_0) \\
&\quad + \int_{x_0}^{x} \int_{y_0}^{y} \int_{x_0}^{u} \int_{y_0}^{v} \frac{\partial^4 g(a,b)}{\partial x^2 \partial y^2} da\, db\, du\, dv \qquad (5)
\end{aligned}
$$

The first row is a constant plus a function of $x$ and a function of $y$ that may be constructed using a bond and options on $x$ and options on $y$. The second row is a product of $x$ and options on $y$ plus a product of $y$ and options on $x$. The third row involves the product. The fourth order integral may be analyzed as follows. For $x > x_0$ and $y > y_0$, we write

$$
\begin{aligned}
&\int_{x_0}^{x} \int_{y_0}^{y} \int_{x_0}^{u} \int_{y_0}^{v} \frac{\partial^4 g}{\partial x^2 \partial y^2}(a,b) da\, db\, du\, dv \\
&= \int_{x_0}^{x} \int_{y_0}^{y} da\, db \int_{a}^{x} \int_{b}^{y} \frac{\partial^4 g}{\partial x^2 \partial y^2}(a,b) du\, dv \qquad (6) \\
&= \int_{x_0}^{x} \int_{y_0}^{y} da\, db \frac{\partial^4 g}{\partial x^2 \partial y^2}(a,b)(x - a)(y - b) \qquad (7) \\
&= \int_{x_0}^{\infty} \int_{y_0}^{\infty} da\, db \frac{\partial^4 g}{\partial x^2 \partial y^2}(a,b)(x - a)^{+}(y - b)^{+} \qquad (8)
\end{aligned}
$$

In the positive orthant relative to the point $x_0, y_0$, one may replicate such functions with positions in product calls at strikes $(a,b)$. In the region $x < x_0$ and $y > y_0$, one may write

$$
\begin{aligned}
&\int_{x}^{x_0} \int_{y_0}^{y} \int_{u}^{x_0} \int_{y_0}^{v} \frac{\partial^4 g}{\partial x^2 \partial y^2}(a,b) da\, db\, du\, dv \\
&= \int_{x}^{x_0} \int_{y_0}^{y} da\, db \int_{x}^{a} \int_{b}^{y} \frac{\partial^4 g}{\partial x^2 \partial y^2}(a,b) du\, dv \qquad (9) \\
&= \int_{x}^{x_0} \int_{y_0}^{y} da\, db \frac{\partial^4 g}{\partial x^2 \partial y^2}(a,b)(a - x)(y - b) \qquad (10) \\
&= \int_{0}^{x_0} \int_{y_0}^{\infty} da\, db \frac{\partial^4 g}{\partial x^2 \partial y^2}(a,b)(a - x)^{+}(y - b)^{+} \qquad (11)
\end{aligned}
$$

In the second orthant relative to $x_0, y_0$ products of puts on $x$ and calls on $y$ are employed. Similarly, in the third orthant, it is products of puts, and, in the fourth, it is calls on $x$ and puts on $y$. The general result may be stated as follows.

$$
\begin{aligned}
g(x,y) \;=\; & g(x_0,y) + g(x,y_0) - g(x_0,y_0) \\
& + (x - x_0)\left[\frac{\partial g(x_0,y)}{\partial x} - \frac{\partial g(x_0,y_0)}{\partial x}\right] + (y - y_0)\left[\frac{\partial g(x,y_0)}{\partial y} - \frac{\partial g(x_0,y_0)}{\partial y}\right] \\
& - \frac{\partial^2 g(x_0,y_0)}{\partial x \partial y}(x - x_0)(y - y_0) \\
& + \int_0^{x_0}\int_0^{y_0} da\,db\, \frac{\partial^4 g}{\partial x^2 \partial y^2}(a,b)(a - x)^+(b - y)^+
\end{aligned}
$$

$$
+ \int_0^{x_0}\int_{y_0}^{\infty} da\,db\, \frac{\partial^4 g}{\partial x^2 \partial y^2}(a,b)(a - x)^+(y - b)^+ \tag{12}
$$

$$
+ \int_{x_0}^{\infty}\int_0^{y_0} da\,db\, \frac{\partial^4 g}{\partial x^2 \partial y^2}(a,b)(x - a)^+(b - y)^+ \tag{13}
$$

$$
+ \int_{x_0}^{\infty}\int_{y_0}^{\infty} da\,db\, \frac{\partial^4 g}{\partial x^2 \partial y^2}(a,b)(x - a)^+(y - b)^+. \tag{14}
$$

A classic application of functional replication in one dimension is the pricing of the variance swap contract or the computation of the VIX index. Other applications include the computation of the CBOE skew index. In two dimensions, applications would include the replication of straddles and strangles on the spread of one asset price over another.

### 3. Multivariate Geometric Brownian Motion and Product Options

Suppose two assets, $S_1, S_2$, are driven by correlated Brownian motions with mean rates of return $\mu_1, \mu_2$ with volatilities $\sigma_1, \sigma_2$ and correlation $\rho$. For a pair of correlated standard Brownian motions $W_1, W_2$, the asset price dynamics under the true or physical measures are

$$
\begin{aligned}
dS_1 &= \mu_1 S_1 dt + \sigma_1 S_1 dW_1 \tag{15} \\
dS_2 &= \mu_2 S_2 dt + \sigma_2 S_2 dW_2 \tag{16} \\
dW_1 dW_2 &= \rho dt \tag{17}
\end{aligned}
$$

Consider a product call paying at maturity for strike $K_1, K_2$ the cash flow $(S_1 - K_1)^+(S_2 - K_2)^+$. Let the value of the option prior to maturity be $C(S_1, S_2, t)$ if, at time $t$, the stock prices are $S_1, S_2$. The differential of the call price is

$$
\begin{aligned}
dC \;=\; & C_t dt + C_{S_1}\mu_1 S_1 dt + C_{S_2}\mu_2 S_2 dt \tag{18} \\
& + \frac{1}{2}C_{S_1 S_1}\sigma_1^2 S_1^2 dt + \frac{1}{2}C_{S_2 S_2}\sigma_2^2 S_2^2 dt + C_{S_1 S_2}\rho\sigma_1\sigma_2 S_1 S_2 dt \tag{19} \\
& + C_{S_1}\sigma_1 S_1 dW_1 + C_{S_2}\sigma_2 S_2 dW_2 \tag{20}
\end{aligned}
$$

Consider a short call that is delta hedged with hedge returns

$$
C_S \mu_1 S dt + C_S \sigma_1 S dW_1 \tag{21}
$$

$$
C_V \mu_2 V dt + C_V \sigma_2 V dW_2. \tag{22}
$$

Then, the hedged short call has a risk free return that must be the interest on value, or

$$
-C_t - \frac{1}{2}C_{S_1 S_1}\sigma_1^2 S_1^2 - \frac{1}{2}C_{S_2 S_2}\sigma_2^2 S_2^2 - C_{S_1 S_2}\rho\sigma_1\sigma_2 S_1 S_2 = r\left(C_{S_1}S_2 + C_{S_2}S_2 - C\right). \tag{23}
$$

Reversing time, the value function is a solution to the equation

$$
C_T = rS_1 C_{S_1} + rS_2 C_{S_2} + \frac{1}{2}C_{S_1 S_1}\sigma_1^2 S_1^2 + \frac{1}{2}C_{S_2 S_2}\sigma_2^2 S_2^2 + C_{S_1 S_2}\rho\sigma_1\sigma_2 S_1 S_2 - rC \tag{24}
$$

subject to the boundary condition

$$C(S, V, T) = (S_1 - K_1)^+ (S_2 - K_2)^+. \tag{25}$$

From the Feynmann–Kac relationship between partial differential equations and expectations, the value is given by

$$C(S_1, S_2, t) = e^{-rt} E\left[ \left( S_1 e^{rt + \sigma_1 \sqrt{t} z_1 - \sigma_1^2 t/2} - K_1 \right)^+ \left( S_2 e^{rt + \sigma_2 \sqrt{t} z_2 - \sigma_2^2 t/2} - K_2 \right)^+ \right], \tag{26}$$

where $(z_1, z_2)$ has a standard bivariate normal distribution with correlation $\rho$. The product call option pricing formula may be developed on solving this integration problem. One may also assert Equation (26) directly by appealing to the existence of a single risk neutral measure; however, for completeness one should also display the explicit measure change. However, the explicit replication strategy is not as clear in such an approach.

The domain of integration is

$$z_1 \quad > \quad d = \frac{\ln(K_1/S_1)}{\sigma_1 \sqrt{t}} - \frac{r}{\sigma_1} \sqrt{t} + \frac{\sigma_1}{2} \sqrt{t} \tag{27}$$

$$z_2 \quad > \quad e = \frac{\ln(K_2/S_2)}{\sigma_2 \sqrt{t}} - \frac{r}{\sigma_2} \sqrt{t} + \frac{\sigma_2}{2} \sqrt{t} \tag{28}$$

The forward product call price $e^{rt} C(S_1, S_2, t)$ is given by the double integral

$$\int_d^\infty \int_e^\infty \left( S_1 e^{rt + \sigma_1 \sqrt{t} z_1 - \sigma_1^2 t/2} - K_1 \right) \left( S_2 e^{rt + \sigma_2 \sqrt{t} z_2 - \sigma_2^2 t/2} - K_2 \right) f(z_1, z_2) dz_1 dz_2 \tag{29}$$

There are four terms and

$$e^{rt} C(S_1, S_2, t) = I_1 - I_2 - I_3 + I_4 \tag{30}$$

where

$$I_1 \quad = \quad S_1 S_2 e^{2rt} \int_d^\infty \int_e^\infty e^{\sigma_1 \sqrt{t} z_1 - \sigma_1^2 t/2} e^{\sigma_2 \sqrt{t} z_2 - \sigma_2^2 t/2} f(z_1, z_2) dz_1 dz_2 \tag{31}$$

$$I_2 \quad = \quad K_1 S_2 e^{rt} \int_d^\infty \int_e^\infty e^{\sigma_2 \sqrt{t} z_2 - \sigma_2^2 t/2} f(z_1, z_2) dz_1 dz_2 \tag{32}$$

$$I_3 \quad = \quad K_2 S_1 e^{rt} \int_d^\infty \int_e^\infty e^{\sigma_1 \sqrt{t} z_1 - \sigma_1^2 t/2} f(z_1, z_2) dz_1 dz_2 \tag{33}$$

$$I_4 \quad = \quad K_1 K_2 \int_d^\infty \int_e^\infty f(z_1, z_2) dz_1 dz_2 \tag{34}$$

Here, $f$ is the bivariate normal density.

We then have that

$$I_4 \quad = \quad 1 - (N(d) + N(e) - bvncdf(d, e, \rho)) \tag{35}$$

$$= \quad \int_d^\infty \frac{1}{\sqrt{2\pi}} e^{-w^2/2} N\left( \frac{\rho w - e}{\sqrt{1 - \rho^2}} \right) dw$$

where $bvncdf$ is the bivariate normal cummulative distribution function.

For $I_3$, we write the joint density as

$$f(z_1, z_2) = \frac{1}{\sqrt{2\pi}} e^{-z_1^2/2} \frac{1}{\sqrt{2\pi(1 - \rho^2)}} e^{-\frac{(z_2 - \rho z_1)^2}{2(1 - \rho^2)}} \tag{36}$$

It follows that

$$
I_3 = K_2 S_1 e^{rt} \int_d^\infty e^{\sigma_1 \sqrt{t} z_1 - \sigma_1^2 t/2} \frac{1}{\sqrt{2\pi}} e^{-z_1^2/2} N\left(\frac{\rho z_1 - e}{\sqrt{1-\rho^2}}\right) \tag{37}
$$

$$
= K_2 S_1 e^{rt} \int_{d-\sigma_1\sqrt{t}}^\infty \frac{1}{\sqrt{2\pi}} e^{-w^2/2} N\left(\frac{\rho(w+\sigma_1\sqrt{t})-e}{\sqrt{1-\rho^2}}\right) dw \tag{38}
$$

Similarly,

$$
I_2 = K_1 S_2 e^{rt} \int_{e-\sigma_2\sqrt{t}}^\infty \frac{1}{\sqrt{2\pi}} e^{-w^2/2} N\left(\frac{\rho(w+\sigma_2\sqrt{t})-d}{\sqrt{1-\rho^2}}\right) dw \tag{39}
$$

Finally,

$$
I_1 = S_1 S_2 e^{2rt} \int_d^\infty \int_e^\infty e^{\sigma_1\sqrt{t}z_1 - \sigma_1^2 t/2} e^{\sigma_2\sqrt{t}z_2 - \sigma_2^2 t/2} \times \tag{40}
$$

$$
\frac{1}{\sqrt{2\pi}} e^{-z_1^2/2} \frac{1}{\sqrt{2\pi(1-\rho^2)}} e^{-\frac{(z_2-\rho z_1)^2}{2(1-\rho^2)}} \tag{41}
$$

$$
= S_1 S_2 e^{2rt} \int_d^\infty e^{\sigma_1\sqrt{t}z_1 - \sigma_1^2 t/2} \frac{1}{\sqrt{2\pi}} e^{-z_1^2/2} dz_1 \times \tag{42}
$$

$$
\int_e^\infty e^{\sigma_2\sqrt{t}z_2 - \sigma_2^2 t/2} \frac{1}{\sqrt{2\pi(1-\rho^2)}} e^{-\frac{(z_2-\rho z_1)^2}{2(1-\rho^2)}} dz_2 \tag{43}
$$

$$
= S_1 S_2 e^{2rt} \int_d^\infty e^{\sigma_1\sqrt{t}z_1 - \sigma_1^2 t/2} \frac{1}{\sqrt{2\pi}} e^{-z_1^2/2} dz_1 \times \tag{44}
$$

$$
\int_{\frac{e-\rho z_1}{\sqrt{1-\rho^2}}}^\infty \frac{1}{\sqrt{2\pi}} e^{-w^2/2} e^{\sigma_2\sqrt{t}(\rho z_1 + \sqrt{1-\rho^2}w) - \sigma_2^2 t/2} dw \tag{45}
$$

$$
= S_1 S_2 e^{2rt} \int_d^\infty e^{\sigma_1\sqrt{t}z_1 - \sigma_1^2 t/2} \frac{1}{\sqrt{2\pi}} e^{-z_1^2/2} e^{\sigma_2\sqrt{t}\rho z_1} \times \tag{46}
$$

$$
\int_{\frac{e-\rho z_1}{\sqrt{1-\rho^2}}}^\infty \frac{1}{\sqrt{2\pi}} e^{-w^2/2 + \sigma_2\sqrt{t}\sqrt{1-\rho^2}w - -\sigma_2^2 t/2} dw \tag{47}
$$

$$
= S_1 S_2 e^{2rt} \int_d^\infty e^{\sigma_1\sqrt{t}z_1 - \sigma_1^2 t/2} \frac{1}{\sqrt{2\pi}} e^{-z_1^2/2} e^{\sigma_2\sqrt{t}\rho z_1} \times \tag{48}
$$

$$
\int_{\frac{e-\rho z_1}{\sqrt{1-\rho^2}}}^\infty \frac{1}{\sqrt{2\pi}} e^{-(w-\sigma_2\sqrt{t}\sqrt{1-\rho^2})^2/2 - \rho^2\sigma_2^2 t/2} dw \tag{49}
$$

$$
= S_1 S_2 e^{2rt} \int_d^\infty e^{\sigma_1\sqrt{t}z_1 - \sigma_1^2 t/2} \times \tag{50}
$$

$$
\frac{1}{\sqrt{2\pi}} e^{-z_1^2/2} e^{\sigma_2\sqrt{t}\rho z_1 - \rho^2\sigma_2^2 t/2} N\left(\frac{\rho z_1 + (1-\rho^2)\sigma_2\sqrt{t} - e}{\sqrt{1-\rho^2}}\right) dz_1 \tag{51}
$$

$$
= S_1 S_2 e^{2rt + \rho\sigma_1\sigma_2 t} \int_{d-\sigma_1\sqrt{t}-\sigma_2\sqrt{t}\rho}^\infty \frac{1}{\sqrt{2\pi}} e^{-w^2/2} \times \tag{52}
$$

$$
N\left(\frac{\rho w + \rho\sigma_1\sqrt{t} + \sigma_2\sqrt{t} - e}{\sqrt{1-\rho^2}}\right) \tag{53}
$$

We, thus, obtain

$$
\begin{aligned}
I_1 &= S_1 S_2 e^{2rt+\rho\sigma_1\sigma_2 t}(1 - (N(d_1) + N(e_1) - bvncdf(d_1, e_1, \rho))) & (54)\\
I_2 &= K_1 S_2 e^{rt}(1 - (N(d_2) + N(e_2) - bvncdf(d_2, e_2, \rho))) & (55)\\
I_3 &= K_2 S_1 e^{rt}(1 - (N(d_3) + N(e_3) - bvncdf(d_3, e_3, \rho))) & (56)\\
I_4 &= K_1 K_2(1 - (N(d) + N(e) - bvncdf(d, e, \rho)) & (57)\\
d_1 &= d - \sigma_1\sqrt{t} - \rho\sigma_2\sqrt{t} & (58)\\
e_1 &= e - \sigma_2\sqrt{t} - \rho\sigma_1\sqrt{t} & (59)\\
d_2 &= d - \rho\sigma_2\sqrt{t} & (60)\\
e_2 &= e - \sigma_2\sqrt{t} & (61)\\
d_3 &= d - \sigma_1\sqrt{t} & (62)\\
e_3 &= e - \rho\sigma_1\sqrt{t} & (63)\\
d &= \frac{\ln(K_1/S_1)}{\sigma_1\sqrt{t}} - \frac{r}{\sigma_1}\sqrt{t} + \frac{\sigma_1}{2}\sqrt{t} & (64)\\
e &= \frac{\ln(K_2/S_2)}{\sigma_2\sqrt{t}} - \frac{r}{\sigma_2}\sqrt{t} + \frac{\sigma_2}{2}\sqrt{t} & (65)
\end{aligned}
$$

The option price is

$$
\begin{aligned}
C(S_1, S_2, t) &= S_1 S_2 e^{rt+\rho\sigma_1\sigma_2 t}(1 - (N(d_1) + N(e_1) - bvncdf(d_1, e_1))) & (66)\\
& \quad - K_1 S_2(1 - (N(d_2) + N(e_2) - bvncdf(d_2, e_2)))\\
& \quad - K_2 S_1(1 - (N(d_3) + N(e_3) - bvncdf(d_3, e_3)))\\
& \quad + K_1 K_2 e^{-rt}(1 - (N(d) + N(e) - bvncdf(d, e)))\\
d &= \frac{\ln(K_1/S_1)}{\sigma_1\sqrt{t}} - \frac{r}{\sigma_1}\sqrt{t} + \frac{\sigma_1}{2}\sqrt{t}\\
e &= \frac{\ln(K_2/S_2)}{\sigma_2\sqrt{t}} - \frac{r}{\sigma_2}\sqrt{t} + \frac{\sigma_2}{2}\sqrt{t}\\
d_1 &= d - \sigma_1\sqrt{t} - \rho\sigma_2\sqrt{t}\\
e_1 &= e - \sigma_2\sqrt{t} - \rho\sigma_1\sqrt{t}\\
d_2 &= d - \rho\sigma_2\sqrt{t}\\
e_2 &= e - \sigma_2\sqrt{t}\\
d_3 &= d - \sigma_1\sqrt{t}\\
e_3 &= e - \rho\sigma_1\sqrt{t}
\end{aligned}
$$

Consider now the product of a put times a call with value

$$
e^{-rt} E\left[\left(K_1 - S_1 e^{rt+\sigma_1\sqrt{t}z_1 - \sigma_1^2 t/2}\right)^+ \left(S_2 e^{rt+\sigma_2\sqrt{t}z_2 - \sigma_2^2 t/2} - K_2\right)^+\right] \quad (67)
$$

The domain of integration is

$$
\begin{aligned}
z_1 &< d = \frac{\ln(K/S)}{\sigma_1\sqrt{t}} - \frac{r}{\sigma_1}\sqrt{t} + \frac{\sigma_1}{2}\sqrt{t} & (68)\\
z_2 &> e & (69)
\end{aligned}
$$

The result is of the form

$$
I_2 - I_1 + I_3 - I_4 \quad (70)
$$

and the integral $I_2$ is

$$K_1 S_2 e^{rt} \int_{-\infty}^{d} \int_{e}^{\infty} e^{\sigma_2 \sqrt{t} z_2 - \sigma_2^2 t/2} f(z_1, z_2) dz_1 dz_2$$

$$= \int_{e-\sigma_2\sqrt{t}}^{\infty} \frac{1}{\sqrt{2\pi}} e^{-w^2/2} N\left( \frac{d - \rho\sigma_2\sqrt{t} - \rho w}{\sqrt{1-\rho^2}} \right) dw.$$

There are four cases denoted $cc, cp, pc,$ and $pp$ for whether the option is a product of two calls, a call and a put, a put and a call, or two puts, respectively. For two calls and two puts, the forward price is $I_1 - I_2 - I_3 + I_4$, and, for cases $cp, pc,$ the forward price is given by $I_2 - I_1 + I_3 - I_4$. In each of the integrals, the probability is a bivariate normal probability of the appropriate quadrant defined by the point $(d, e)$.

## 4. Fast Fourier Transform for Product Options

For a joint density $f(x, y)$ of a pair of log returns, define the product call price by

$$w(a, b) = \int_{a}^{\infty} \int_{b}^{\infty} (e^x - e^a)\left(e^y - e^b\right) f(x, y) dx dy \tag{71}$$

for a joint density $f(x, y)$.

Further suppose the joint characteristic function $\phi(u, v)$ of the joint density is available in closed form. By definition,

$$\phi(u, v) = \int_{-\infty}^{\infty} \int_{-\infty}^{\infty} e^{iux+ivy} f(x, y) dx dy. \tag{72}$$

Following the methods of Carr and Madan (1999), define

$$\gamma(a, b) = \int_{-\infty}^{\infty} \int_{-\infty}^{\infty} e^{\alpha a + \beta b + iua + ivb} w(a, b) da db \tag{73}$$

$$= \int_{-\infty}^{\infty} \int_{-\infty}^{\infty} e^{i(u-i\alpha)a + i(v-i\beta)b} \times \tag{74}$$

$$\int_{a}^{\infty} \int_{b}^{\infty} (e^x - e^a)\left(e^y - e^b\right) f(x, y) dx dy da db \tag{75}$$

$$= \int_{-\infty}^{\infty} \int_{-\infty}^{\infty} f(x, y) dx dy \times \tag{76}$$

$$\int_{-\infty}^{x} \int_{-\infty}^{y} e^{i(u-i\alpha)a + i(v-i\beta)} (e^x - e^a)\left(e^y - e^b\right) da db \tag{77}$$

$$= \int_{-\infty}^{\infty} \int_{-\infty}^{\infty} f(x, y) dx dy \times \int_{-\infty}^{x} \int_{-\infty}^{y} da db \tag{78}$$

$$\left( \begin{array}{c} e^{i(u-i\alpha)a + i(v-i\beta)+x+y} - e^{i(u-i(\alpha+1))a + i(v-i\beta)b+y} - \\ e^{i(u-i\alpha)a + i(v-i(\beta+1))b+x} + e^{i(u-i(\alpha+1))a + i(v-i(\beta+1))b} \end{array} \right) \tag{79}$$

$$= \phi(u - i(\alpha+1), v - i(\beta+1)) \times \tag{80}$$

$$\left( \begin{array}{c} \frac{-1}{(u-i\alpha)(v-i\beta)} + \frac{1}{(u-i(\alpha+1))(v-i\beta)} + \\ \frac{1}{(u-i\alpha)(v-i(\beta+1))} - \frac{1}{(u-i(\alpha+1))(v-i(\beta+1))} \end{array} \right) \tag{81}$$

$$= \frac{\phi(u - i(\alpha+1), v - i(\beta+1))}{\left((u-i\alpha)^2 - i(u-i\alpha)\right)\left((v-i\beta)^2 - i(v-i\beta)\right)} \tag{82}$$

It follows that

$$w(a, b) = \frac{1}{4\pi^2} \int_{-\infty}^{\infty} \int_{-\infty}^{\infty} e^{-iua-ivb} \frac{e^{-\alpha a - \beta b} \phi(u - i(\alpha+1), v - i(\beta+1))}{\left((u-i\alpha)^2 - i(u-i\alpha)\right)\left((v-i\beta)^2 - i(v-i\beta)\right)} du dv \tag{83}$$

Hence, by two-dimensional fast Fourier inversion, we may price product options given the joint characteristic function.

For the product of put and a call defined for $\alpha > 1, \beta > 0$

$$\gamma_{pc}(a,b) = \int_{-\infty}^{\infty} \int_{-\infty}^{\infty} e^{i(u+i\alpha)a} e^{i(v-i\beta)b} w_{pc}(a,b) da db \tag{84}$$

$$= \int_{-\infty}^{\infty} \int_{-\infty}^{\infty} \int_{-\infty}^{a} \int_{b}^{\infty} e^{i(u+i\alpha)a} e^{i(v-i\beta)} \times \tag{85}$$

$$\left(e^a - e^x\right)\left(e^y - e^b\right) f(x,y) dx dy da db \tag{86}$$

$$= \int_{-\infty}^{\infty} \int_{-\infty}^{\infty} f(x,y) dx dy \int_{x}^{\infty} \int_{-\infty}^{y} \times \tag{87}$$

$$e^{i(u+i\alpha)a} e^{i(v-i\beta)b} \left(e^a - e^x\right)\left(e^y - e^b\right) da db \tag{88}$$

$$= \phi(u + i(\alpha - 1), v - i(\beta + 1)) \times \tag{89}$$

$$\left[ \begin{array}{c} \frac{1}{(u+i(\alpha-1))(v-i\beta)} - \frac{1}{(u+i(\alpha-1))(v-i(\beta+1))} \\ - \frac{1}{(u+i\alpha)(v-i\beta)} + \frac{1}{(u+i\alpha)(v-i(\beta+1))} \end{array} \right] \tag{90}$$

$$= \frac{\phi(u + i(\alpha - 1), v - i(\beta + 1))}{\left((u+i\alpha)^2 - i(u+i\alpha)\right)\left((v-i\beta)^2 - i(v-i\beta)\right)} \tag{91}$$

Similarly, we will have

$$\gamma_{cp}(a,b) = \frac{\phi(u - i(\alpha + 1), v + i(\beta - 1))}{\left((u-i\alpha)^2 - i(u-i\alpha)\right)\left((u+i\beta)^2 - i(u+i\beta)\right)} \tag{92}$$

$$\gamma_{pp}(a,b) = \frac{\phi(u + i(\alpha - 1), v + i(\beta - 1))}{\left((u+i\alpha)^2 - i(u+i\alpha)\right)\left((u+i\beta)^2 - i(u+i\beta)\right)}. \tag{93}$$

It follows that

$$w_{cc}(a,b) = \frac{1}{4\pi^2} \int_{-\infty}^{\infty} \int_{-\infty}^{\infty} e^{-iua-ivb} \times \tag{94}$$

$$\frac{e^{-\alpha a - \beta b} \phi(u - i(\alpha + 1), v - i(\beta + 1))}{\left((u-i\alpha)^2 - i(u-i\alpha)\right)\left((v-i\beta)^2 - i(v-i\beta)\right)} da db \tag{95}$$

$$w_{pc}(a,b) = \frac{1}{4\pi^2} \int_{-\infty}^{\infty} \int_{-\infty}^{\infty} e^{-iua-ivb} \times \tag{96}$$

$$\frac{e^{\alpha a - \beta b} \phi(u + i(\alpha - 1), v - i(\beta + 1))}{\left((u+i\alpha)^2 - i(u+i\alpha)\right)\left((v-i\beta)^2 - i(v-i\beta)\right)} da db \tag{97}$$

$$w_{cp}(a,b) = \frac{1}{4\pi^2} \int_{-\infty}^{\infty} \int_{-\infty}^{\infty} e^{-iua-ivb} \times \tag{98}$$

$$\frac{e^{-\alpha a + \beta b} \phi(u - i(\alpha + 1), v + i(\beta - 1))}{\left((u-i\alpha)^2 - i(u-i\alpha)\right)\left((u+i\beta)^2 - i(u+i\beta)\right)} da db \tag{99}$$

$$w_{pp}(a,b) = \frac{1}{4\pi^2} \int_{-\infty}^{\infty} \int_{-\infty}^{\infty} e^{-iua-ivb} \times \tag{100}$$

$$\frac{e^{\alpha a + \beta b} \phi(u + i(\alpha - 1), v + i(\beta - 1))}{\left((u+i\alpha)^2 - i(u+i\alpha)\right)\left((u+i\beta)^2 - i(u+i\beta)\right)} da db \tag{101}$$

For puts, the values of $\alpha, \beta$ need to be above unity.

The inversion may be accomplished using a two dimensional fast Fourier Transform. The fast Fourier transform $Y$ of an $m \times n$ matrix $X$ evaluates the double sum

$$Y_{p+1,q+1} = \sum_{j=0}^{m-1} \sum_{k=0}^{n-1} \omega_m^{jp} \omega_n^{kq} X_{j+1,k+1}, \; p = 0, \cdots m-1, \; q = 0, \cdots n-1 \tag{102}$$

where

$$\omega_m = e^{-2\pi i/m} \tag{103}$$
$$\omega_n = e^{-2\pi i/n} \tag{104}$$

The integral to be evaluated is

$$\int_{-\infty}^{\infty} \int_{-\infty}^{\infty} e^{-iua} e^{-ivb} g(u,v) \, du \, dv \tag{105}$$

We may approximate by

$$\int_{-M}^{M} \int_{-N}^{N} e^{-iua-ivb} g(u,v) \, du \, dv \tag{106}$$

We discretize with spacing $\lambda_u, \lambda_v$

$$u_j = -M + \lambda_u j \tag{107}$$
$$v_k = -N + \lambda_v k \tag{108}$$

We also discretize in strike space with spacing $\eta_a, \eta_b$ by

$$a_p = -A + \eta_a p \tag{109}$$
$$b_q = -B + \eta_b q \tag{110}$$

to approximate the double integral by the double sum

$$W(a_p, b_q) = \sum_{j=0}^{m-1} \sum_{k=0}^{n-1} e^{-i(-M+\lambda_u j)(-A+\eta_a p)} e^{-i(-N+\lambda_v k)(-B+\eta_b q)} \tag{111}$$

$$\times g(-M + \lambda_u j, -N + \lambda_v k) \lambda_u \lambda_v \tag{112}$$

$$e^{-iM(-A+\eta_a p)} W(a_p, b_q) e^{-iN((-B+\eta_b q)} \tag{113}$$

$$= \sum_{j=0}^{m-1} \sum_{k=0}^{n-1} e^{-i\lambda_u \eta_a j p} e^{-i\lambda_v \eta_b k q} \times \tag{114}$$

$$\left( \begin{array}{c} e^{iA(-M+\lambda_u j)} \\ \times g(-M + \lambda_u j, -N + \lambda_v k) \\ \times e^{iB(-N+\lambda_v k)} \end{array} \right) \lambda_u \lambda_v \tag{115}$$

Now, we set

$$\lambda_u \eta_a = \frac{2\pi}{m} \tag{116}$$
$$\lambda_v \eta_b = \frac{2\pi}{n} \tag{117}$$

Define

$$X_{j+1,k+1} = e^{iA(-M+\lambda_u j)} g(-M + \lambda_u j, -N + \lambda_v k) e^{iB(-N+\lambda_v k)} \lambda_u \lambda_v \tag{118}$$

Then, define $Y$ via Equation (102). The option prices are obtained as

$$W(a_p, b_q) = e^{iM(-A+\eta_a p)} Y_{p+1,q+1} e^{iN((-B+\eta_b q)}. \tag{119}$$

## 5. Multivariate Bilateral Gamma

An application of the two dimensional fast Fourier transform on pricing product options requires a closed form multivariate characteristic function. The marginals of this

joint characteristic function should be capable of calibrating risk neutral densities observed in the separate option markets for a given maturity. There are a number of pure jump Lévy processes that will fit marginal option prices.

Among them are the three parameter variance gamma model of Madan and Seneta (1990), Madan et al. (1998) and its four parameter extension to the bilateral gamma model proposed in Küchler and Tappe (2008). Recently, Madan and Schoutens (2020), Madan (2020), and Madan and Wang (2020a) have investigated multivariate models with closed form characteristic functions consistent with arbitrary marginal bilateral gamma models. Here, we present a summary of the required characteristic functions.

Consider first the bilateral gamma process for the marginals. Let $(\gamma_p(t), t > 0)$, $(\gamma_n(t), t > 0)$ be two independent standard gamma processes with unit mean and variance rates. Now, we introduce four parameters $b_p, c_p, b_n, c_n$ representing the scale and speed of positive and negative moves, respectively. The bilateral gamma process $X_{BG}(t)$ is given by

$$X_{BG}(t) = b_p \gamma_p(c_p t) - b_n \gamma_n(c_n t). \tag{120}$$

The process is a Lévy process of independent and identically distributed increments. It is also a pure jump process with the characteristic function

$$E[\exp iu X_{BG}(t)] = \left(\frac{1}{1 - iub_p}\right)^{c_p t} \left(\frac{1}{1 + iub_n}\right)^{c_n t}. \tag{121}$$

The characteristic function has a Lévy–Khintchine decomposition in terms of a Lévy density $k(x)$ that announces the arrival rate of jumps of size $x$ as follows.

$$E[\exp iu X_{BG}(t)] = \exp\left(\int_{-\infty}^{\infty} \left(e^{iux} - 1\right) k(x) dx\right). \tag{122}$$

The specific form for the Lévy density is

$$k(x) = c_p \frac{\exp\left(-\frac{x}{b_p}\right)}{|x|} \mathbf{1}_{x>0} + c_n \frac{\exp\left(-\frac{|x|}{b_n}\right)}{|x|} \mathbf{1}_{x<0}. \tag{123}$$

The density for any horizon may be obtained by Fourier inversion of the characteristic function using the Fast Fourier Transform. The density also has a closed form in terms of the Whittaker W function (Küchler and Tappe (2008)). We denote this function by $W(a, b)$. For $x > 0$, the density is given by

$$
\begin{aligned}
f_{BG}(x) &= \left(\frac{1}{b_p}\right)^{c_p} \left(\frac{1}{b_n}\right)^{c_n} \left(\frac{1}{(1/b_p + 1/b_n)^{(c_p+c_n)/2} \Gamma(c_p)}\right) \times \\
&\quad x^{(c_p+c_n)/2-1} \exp\left(-x/2(1/b_p + 1/b_n)\right) \times \\
&\quad W\left(\frac{c_p - c_n}{2}, \frac{c_p + c_n - 1}{2}, \left(\frac{1}{b_p} + \frac{1}{b_n}\right) x\right).
\end{aligned} \tag{124}
$$

For $x < 0$, the roles of $b_p, c_p$ and $b_n, c_n$ are reversed.

The multivariate bilateral gamma model is presented in Madan and Schoutens (2020). The presentation follows Buchmann et al. (2019) where a multivariate Lévy process is constructed having a Lévy density with full support on $\mathbb{R}^n - \{0\}$ that is also consistent with prespecified variance gamma marginals displaying different kurtosis levels for each component. The construction in Madan and Schoutens (2020) extends that of Buchmann et al. (2019) to attain consistency with prespecified bilateral gamma marginals.

Let the marginal distributions be bilateral gamma with the parameters $b_{pi}, c_{pi}, b_{ni}$, and $c_{ni}$ for component $i$. The multivariate bilateral gamma model is the sum of a multivariate variance gamma plus independent bilateral gamma shocks. The multivariate variance

gamma is a multivariate normal with the mean vector $\theta$ and covariance $\Sigma$, both of which are scaled by a single gamma variate $g$ with a unit mean and variance $v$. The multivariate variance gamma random vector $X_{MVG}$ may then be written as

$$X_{MVG} = \theta g + \sqrt{g}\sqrt{diag(\Sigma)}Z \tag{125}$$

where the vector $Z = (Z_i, i = 1, \cdots, n)$ is multivariate normal with zero means, unit variances, and correlation matrix $C$. The covariance matrix is

$$\Sigma = \sqrt{diag(\Sigma)}C\sqrt{diag(\Sigma)}. \tag{126}$$

In addition, there are independent bilateral gamma variates $Y_i$ with the parameters $\widetilde{b}_{pi}, \widetilde{c}_{pi}, \widetilde{b}_{ni}$, and $\widetilde{c}_{ni}$ and with $Y = (Y_1, \cdots, Y_n)'$

$$X_{MBG} = X_{MVG} + Y. \tag{127}$$

The only dependency parameters are the correlation matrix $C$ and the variance $v$ of the gamma variate $g$. All other parameters are derived from the parameters of the marginal processes. Specifically, we have that

$$\theta_i = \frac{b_{pi} - b_{ni}}{v} \tag{128}$$

$$\Sigma_{ii} = \frac{2b_{pi}b_{ni}}{v} \tag{129}$$

$$\widetilde{b}_{pi} = b_{pi} \tag{130}$$

$$\widetilde{b}_{ni} = b_{ni} \tag{131}$$

$$\widetilde{c}_{pi} = c_{pi} - \frac{1}{v} \tag{132}$$

$$\widetilde{c}_{ni} = c_{ni} - \frac{1}{v} \tag{133}$$

The parameter $v$ has a lower bound of the reciprocal of the minimum of all the marginal speed parameters $\min_i(\min(c_{pi}, c_{ni}))$.

The characteristic function of the multivariate bilateral gamma vector is given by

$$
\begin{aligned}
\phi_{MBG}(u) &= E\left[\exp(iu'X_{MBG})\right] \\
&= \left(\frac{1}{1 - iu'\theta v + \frac{v}{2}u'\Sigma u}\right)^{\frac{1}{v}}\prod_j\left(\frac{1}{1 - iu_j b_{pj}}\right)^{c_{pj}-1/v}\left(\frac{1}{1 + iu b_{nj}}\right)^{c_{nj}-1/v}.
\end{aligned} \tag{134}
$$

The multivariate arrival rates or Lévy density $k(x)$ for the multivariate bilateral gamma model may be specified as follows.

$$k(x) = \widetilde{m}(x) + \sum_{j=1}^{n}k_j(x_j)\prod_{\substack{l\neq j \\ l=1}}^{M}\mathbf{1}_{x_l=0}, \tag{135}$$

where

$$\widetilde{m}(z) \quad = \quad \frac{\exp\left(\theta^T \Sigma^{-1} x\right)}{\nu(2\pi)^{n/2-1}\sqrt{|\Sigma|}\sqrt{x^T \Sigma^{-1} x}} \times \tag{136}$$

$$\exp\left(-\sqrt{\left(\theta^T \Sigma^{-1}\theta + \frac{2}{\nu}\right)\left(x^T \Sigma^{-1} x\right)}\right)$$

$$k_j(x_j) \quad = \quad \frac{c_{nj} - 1/\nu}{|x_j|}\exp\left(-|x_j|/b_{nj}\right)\mathbf{1}_{x_j<0} \tag{137}$$

$$+ \frac{c_{pj} - 1/\nu}{x_j}\exp\left(-x_j/b_{pj}\right)\mathbf{1}_{x_j>0}. \tag{138}$$

For option pricing, one must incorporate the marginal convexity corrections to match the forward prices for the two assets. This is the characteristic function $\phi_{BGMP}(u)$, and, in the two dimensional case, it is given by

$$\phi_{BGMP}(u) \quad = \quad \phi_{MBG}(u)\exp(iu_1\omega_1 + iu_2\omega_2) \tag{139}$$

$$\omega_1 \quad = \quad \ln(S_1) + (r - q_1)t - \ln\phi_{MBG}((-i,0)) \tag{140}$$

$$\omega_2 \quad = \quad \ln(S_2) + (r - q_2)t - \ln\phi_{MBG}((0,-i)). \tag{141}$$

## 6. Sample Computations Using the Multivariate Bilateral Gamma Model

For an example on product option pricing, we consider product options on *JPM* and *SPY*, both of which have many options trading. On 12 December 2019 for maturities below three months and moneyness, measured by the absolute value of the logarithm of strike to the spot price, below 0.3, there were 226 options trading on *JPM* and 1582 options on *SPY*. The number of days to maturity of traded maturities closest to a month was 29 days. The maturity of 29 days had 36 strikes for *JPM* and 74 strikes for *SPY*. These options may be employed to determine the marginal risk neutral distribution on *JPM* and *SPY* for the 29 day maturity. The marginal bilateral gamma parameters are presented in Table 1.

**Table 1.** Marginal BG parameters for the 29 day maturity on 12 December 2019.

| Asset | $b_p$ | $c_p$ | $b_n$ | $c_n$ | $\omega$ |
|-------|-------|-------|-------|-------|----------|
| JPM | 0.0241 | 18.2249 | 0.0398 | 16.8810 | 0.2142 |
| SPY | 0.0270 | 2.2784 | 0.0509 | 6.6806 | 0.2693 |

Figure 1 presents a graph of the observed and model option prices for the 29 day maturity on 12 December 2019 for the two assets.

The two dependency parameters were set as follows. The value of $\nu = 0.4828$ was ten percent above the lower bound and the correlation was set at 0.6. For this parameter setting and the spot for both assets set to 100 with strikes ranging between 80 and 120 Figure 2 presents four graphs for product option prices computed by two dimensional fast Fourier inversion of the Fourier transform of modified product option prices as per Equations (82), (91), (92), and (93).

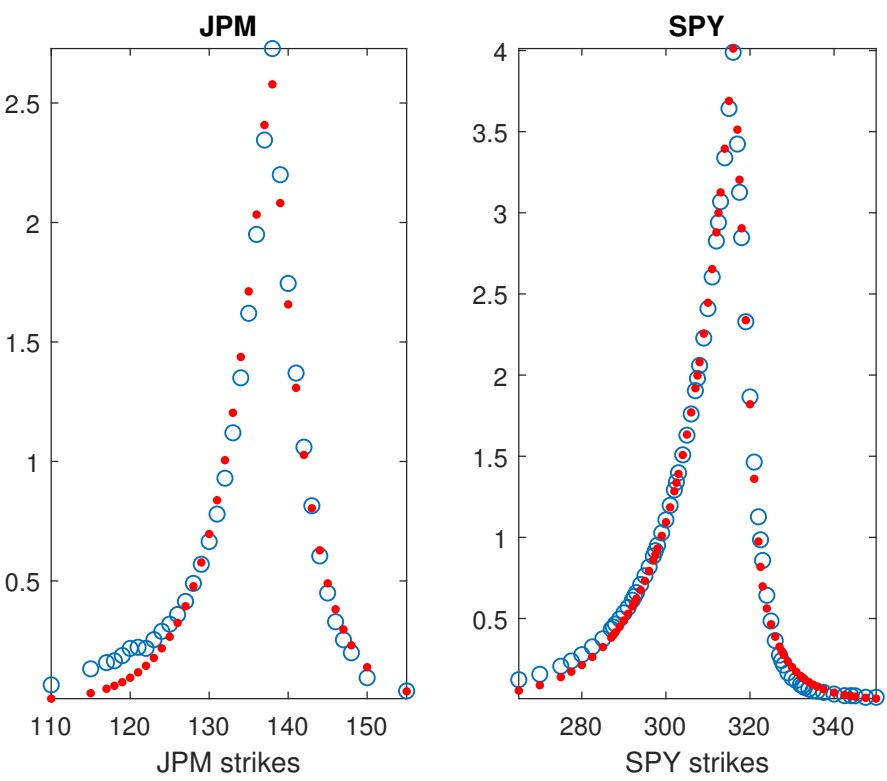

**Figure 1.** Bilateral Gamma marginals fit to JPM and SPY options for a maturity of 29 days on 12 December 2019. The circles represent marlet prices. Model prices are shown by dots.

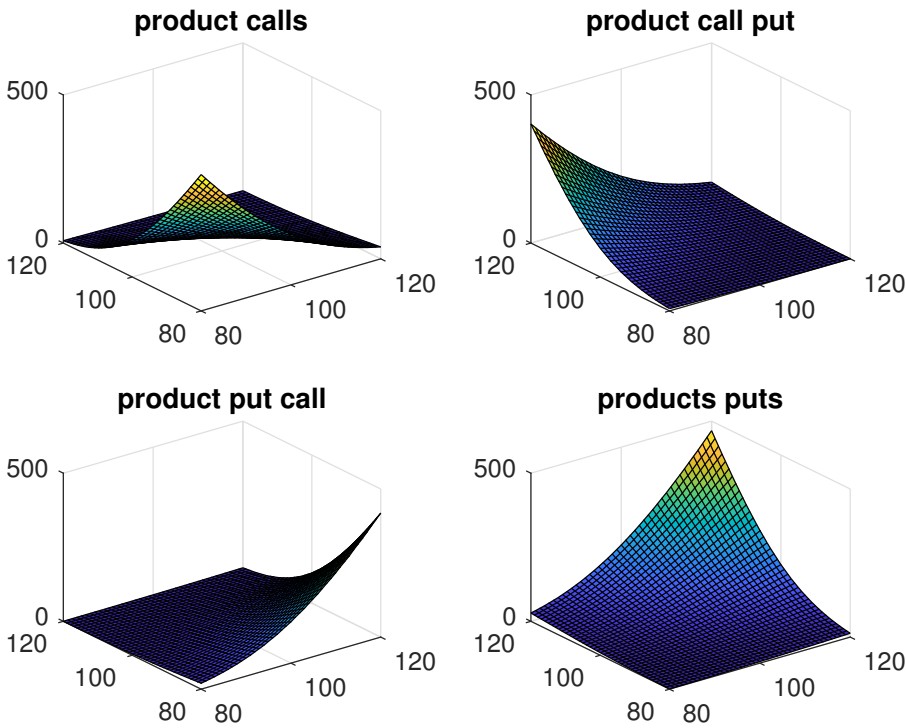

**Figure 2.** Product option prices. The upper panel is products of calls on JPM with calls and puts on SPY. The lower panel is for products of puts on JPM with calls and puts on SPY.

With a view toward studying the effects of dependency parameters on product option prices, we considered four product options with strikes five percent out of the money. The options were a product of two calls, a call and a put, a put and a call, and two puts. There were five settings for correlation from $-0.8$ to $0.8$ in steps of $0.4$. There were four settings for the percentage excess of $\nu$ above its lower bound of $0.25$, $0.5$, $0.75$, and $1.0$. These results are presented in Tables 2–5, for the four option types.

**Table 2.** Product of two calls.

|  | Correlation | | | | |
|---|---|---|---|---|---|
| $\nu$ | $-0.8$ | $-0.4$ | 0 | 0.4 | 0.8 |
| 0.25 | 55.47 | 64.67 | 75.08 | 86.75 | 99.72 |
| 0.5 | 58.35 | 66.15 | 74.85 | 84.48 | 95.08 |
| 0.75 | 60.44 | 67.21 | 74.69 | 82.89 | 91.84 |
| 1.0 | 62.04 | 68.02 | 74.57 | 81.71 | 89.46 |

For the product of two calls, the option price rises with correlation. The effect of $\nu$ is positive for negative correlation, negative for positive correlation, and flat for zero correlation.

**Table 3.** Product of a call and a put.

|  | Correlation | | | | |
|---|---|---|---|---|---|
| $\nu$ | $-0.8$ | $-0.4$ | 0 | 0.4 | 0.8 |
| 0.25 | 98.06 | 86.33 | 75.49 | 65.55 | 56.52 |
| 0.5 | 94.82 | 85.13 | 76.11 | 67.76 | 60.08 |
| 0.75 | 92.52 | 84.28 | 76.55 | 69.35 | 62.68 |
| 1.0 | 90.82 | 83.64 | 76.88 | 70.55 | 64.65 |

For the product of a call and a put, the option price falls with correlation. The effect of $\nu$ is negative for negative correlation, positive for positive correlation, and flat for zero correlation.

**Table 4.** Product of a put and a call.

|  | Correlation | | | | |
|---|---|---|---|---|---|
| $\nu$ | $-0.8$ | $-0.4$ | 0 | 0.4 | 0.8 |
| 0.25 | 90.90 | 80.32 | 70.53 | 61.53 | 53.35 |
| 0.5 | 87.80 | 79.06 | 70.92 | 63.37 | 56.41 |
| 0.75 | 85.61 | 78.18 | 71.20 | 64.69 | 58.65 |
| 1.0 | 83.98 | 77.51 | 71.41 | 65.69 | 60.35 |

For the product of a put and a call, the option price falls with correlation. The effect of $\nu$ is negative for negative correlation, positive for positive correlation, and flat for zero correlation.

**Table 5.** Product of two puts.

|  | Correlation | | | | |
|---|---|---|---|---|---|
| $\nu$ | $-0.8$ | $-0.4$ | 0 | 0.4 | 0.8 |
| 0.25 | 62.59 | 71.39 | 80.77 | 90.72 | 101.3 |
| 0.5 | 64.47 | 71.86 | 79.68 | 87.92 | 96.60 |
| 0.75 | 65.82 | 72.19 | 78.89 | 85.94 | 93.31 |
| 1.0 | 66.85 | 72.44 | 78.31 | 84.45 | 90.87 |

For the product of two puts, the option price rises with correlation. The effect of $\nu$ is positive for negative correlation, negative for positive correlation, and flat for zero correlation.

### 7. Pricing Product Options Using the Physical Measure

For product options, one has the ability to partially hedge the risk using options on the two underlying assets. The prices of options on these assets are informative on the terms at which the product options could trade. However, as the hedge is not perfect, there is residual risk. Recently, Madan and Wang (2020b) reported on an investigation of option pricing at the cost of a hedge plus a charge for residual risk to levels of risk acceptability. For acceptable risk defined by concave distortions of probability as proposed in Cherny and Madan (2009) following Artzner et al. (1999) and Kusuoka (2001), the residual risk charge is just a distorted expectation of the simulated residual risk taken at an appropriate stress level for the distortion employed. For further details, we recommend, apart from the cited papers, Madan and Schoutens (2016). Here, we use the distortion *minmaxvar* with stress parameter $\gamma$ and definition

$$\Psi^{(\gamma)}(u) = 1 - \left(1 - u^{\frac{1}{1+\gamma}}\right)^{1+\gamma}.$$

For the physical measure, we bootstrap pairs of returns from the data immediately prior to the pricing date for the option maturity of interest. Bilateral gamma marginals are fit to this data. The dependency parameters of the multivariate bilateral gamma model are estimated by matching the two dimensional empirical characteristic function to the theoretical counterpart.

The estimated parameters are then used to simulate a hundred thousand readings on pairs of returns. On the simulated space, a hedge is determined by determining the funds needed to cover the residual risk to a level of risk acceptability. This magnitude is the residual risk charge. The cost of the hedge is then obtained from the market prices of the hedging instruments. The product option is priced at the cost of the hedge plus the residual risk charge. This procedure is illustrated on a set of strikes for all four types of product options on data for *XLE* and *XLP* as on 12 December 2019.

The first step is the formulation of joint returns on the two assets to the option maturity. We work with an option maturity of a month or 21 business days. However, an exact 21 days may not be a relevant way of forming joint returns as some months may be longer and others shorter in terms of economic time. We, therefore, take the number of days to be random with a gamma distribution with a mean of 21 days. For the shape parameter, we take the value of 2 that places the mode at half the mean.

The number of days $n$ is simulated as one plus the integer part of the gamma variate. For the random number of days, we randomly select a start date within the last thousand days and an end date at the start date plus $n$. The returns on the two assets between the start and end dates delivers a single reading on joint returns. The procedure is repeated 500 times to draw a sample of 500 monthly joint returns.

In modeling the joint returns, we follow Madan and Wang (2020b). First we lock in a regression of the first asset return on the second and for the pair *XLE* and *XLP*, we obtain the results

$$r_{1n} = -0.0018 + 0.4006 r_{2n} + u_n$$

The $t - statistics$ for this regression were $-0.92$, and $6.38$. The $R - square$ was 6.22 percent.

We then model $r_{2n}$ and $u_n$ to be multivariate bilateral gamma. The marginal bilateral gamma parameters for $r_2$ and $u$ are displayed in Table 6.

**Table 6.** Marginal bilateral gamma parameters.

|       | $b_p$  | $c_p$  | $b_n$  | $c_n$  |
|-------|--------|--------|--------|--------|
| $r_2$ | 0.0134 | 2.7719 | 0.0111 | 2.7719 |
| $u$   | 0.0361 | 1.0546 | 0.0359 | 1.0547 |

The dependency parameters $\nu$ and $\rho$ were estimated by matching the empirical joint characteristic functions at the values 0.9482 and $-0.4769$, respectively. The drifts in the Brownian motions to be time changed were 0.0025 and 0.0003, respectively, for $r_2$ and $u$. The corresponding volatilities were 0.0177 and 0.0523. Setting the initial price levels for the two assets as they were at 31 December 2019 at 60.03 and 62.95 for *XLE* and *XLP*, respectively, we simulated 100, 000 readings for the two asset prices a month later from this joint law for the two returns.

Consider first a product call at the strikes 61.21 and 64 for *XLE* and *XLP*, respectively. The payout on the product call is

$$c(S_1, S_2) = (S_1 - 61.21)^+ (S_2 - 64)^+.$$

By taking positions in the assets and options on the two assets one may access option payouts from the markets of $g(S_1)$ and $h(S_2)$. These functions may be built up from payouts to the assets and option on the assets as a function of the positions taken. One may construct matrices $G$, and $H$ that evaluate the payout on the assets and the options at the 100, 000 readings on the two assets. $G$ and $H$ are then 100, 000 times $n_1, n_2$ respectively, where $n_1, n_2$ are the number of assets in the two markets. With positions given by vectors $a_1, a_2$ of dimensions $n_1, n_2$, we have a hundred thousand readings on hedge cash flows as

$$
\begin{aligned}
g(S_1) &= Ga_1 \\
h(S_2) &= Ha_2.
\end{aligned}
$$

Evaluating the product option payout at the hundred thousand pairs of outcomes the product option payout is also a vector $c(S_1, S_2)$ of length a hundred thousand.

We now propose the construction of the hedge or the positions $a_1, a_2$ in the two sets of assets. We took for hedging assets, the two underlying assets and options at moneyness multiples of a percentage point between plus or minus 10 percent to get 19 and 12 hedging strikes on *XLE* and *XLP*, respectively. The post hedge residual cash flow is

$$R(S_1, S_2) = c(S_1, S_2) - g(S_1) - h(S_2).$$

The residual is the post hedge shortfall on the product option liability and it is to be valued at its ask or upper price, which is the risk charge for holding the residual risk.

The lower price for a risk using distorted expectations is given by its expectation evaluated under a probability distortion. The distorted probability distribution for a risk $X$ with distribution function $F_X(x)$ is $\Psi(F_X(x))$ for a concave distortion $\Psi(u)$, $0 \leq u \leq 1$. The upper price is just the negative of distortion expectation for the negated risk. Defining the distorted expectation by

$$\mathcal{E}(X) = \int_{-\infty}^{\infty} x d\Psi(F_X(x))$$

the upper price or risk charge is

$$rc = -\mathcal{E}(-X).$$

Positions in the hedging assets are found by minimizing the residual risk charge $rc$.

The distortion employed, as already stated, is *minmaxvar*, and we need to pick the stress level $\gamma$ for the distortion. As distorted expectations of hedge fund returns turn negative at stress levels of 0.75, indicating acceptability at the margin for this stress level, we employed the stress level of $\gamma = 0.75$. In constructing hedges, it is imperative to keep the hedging assets neutral, and this was done by centering their cash flows using the

sample mean of the payouts for each asset. The minimal risk charge was 1.9899. Figure 3 displays the hedge cash flows accessed on the two assets. The cost of the hedge was 1.2843, and the price for the product call was 3.2743.

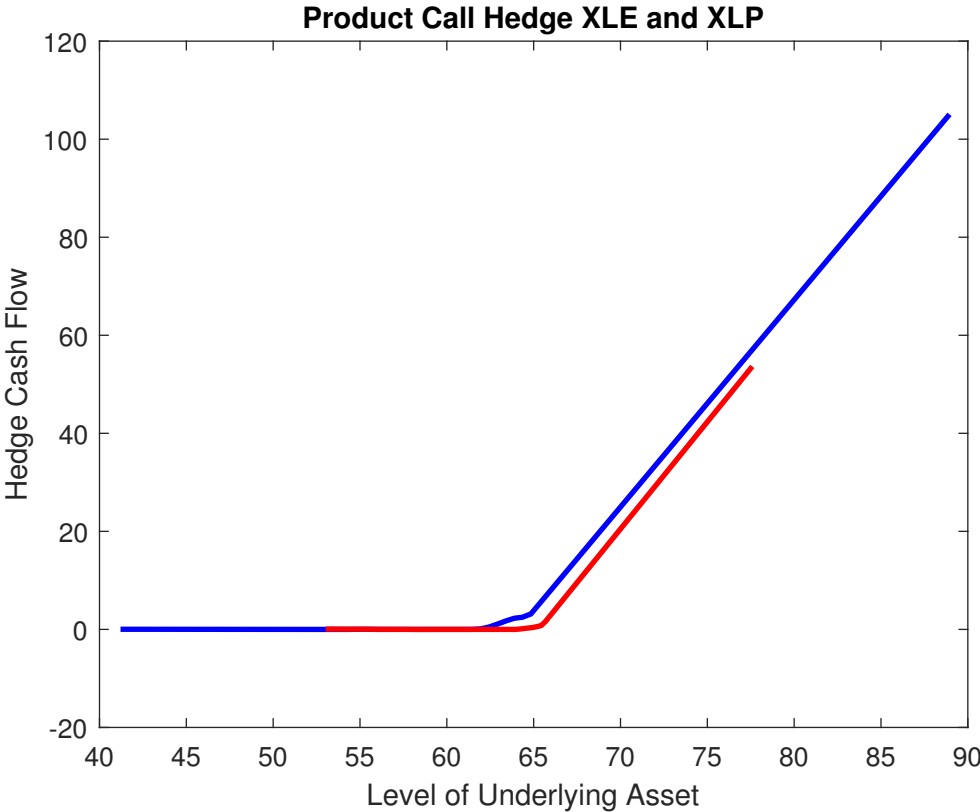

**Figure 3.** Hedge cash flows accessed by risk charge minimizing hedge. The blue curve displays the XLE hedge, and the red curve presents the XLP hedge. The stress level used was 0.75 for the distortion minmaxvar.

With a view to reporting further on the quality of the hedge, we present the percentiles for the hedge cash flows, inclusive of the risk charge, when the payout on the product call is zero. We also present, in Table 7, the percentiles of the shortfall on the product option payout, again inclusive of the risk charge.

**Table 7.** Short fall and zero payout percentiles.

|  | 1 | 5 | 10 | 25 | 50 | 75 | 90 | 95 | 99 |
|---|---|---|---|---|---|---|---|---|---|
| shortfall | −20.78 | −11.06 | −7.24 | −2.81 | −0.71 | −0.22 | 0.11 | 0.30 | 1.03 |
| zero payout | 0.22 | 0.23 | 0.23 | 0.23 | 0.24 | 0.70 | 3.47 | 8.34 | 20.95 |

Table 8 presents the strikes, risk charges, hedge costs, and product option prices for other types of product options hedged using risk charge minimizing hedges.

**Table 8.** Other product options.

| Type | $K_1$ | $K_2$ | *rc* | *hc* | *POP* |
|---|---|---|---|---|---|
| PC | 57.21 | 63 | 1.2009 | 1.4320 | 2.6329 |
| PP | 60.21 | 61 | 0.7096 | 2.9561 | 3.6657 |
| CP | 62.21 | 62.5 | 1.1374 | 0.2 | 1.3374 |

Three types are *PC*, *PP*, and *CP* for products of a put and a call, two puts, and a call and a put. *rc*, *hc*, and *POP* are the risk charge, hedge cost, and the product option price, respectively.

## 8. Conclusions

Product options paying the product of put and or call options payouts at different strikes on two underlying assets were shown to be useful in synthesizing joint densities and replicating sufficiently differentiable functions of two underlying assets. Theory for pricing such options was presented and implemented from three perspectives. The first employed assumptions on asset log returns being jointly driven by correlated Brownian motion. The second used the two dimensional inversion of joint characteristic functions. The third was based on using the physical measure and proposing a risk charge minimizing hedge-using options on the two underlying assets. The options were then priced at the cost of the hedge plus the risk charge for the residual or unhedged risk.

**Author Contributions:** Both authors contributed equally to this article. Both authors have read and agreed to the published version of the manuscript.

**Funding:** This research received no external funding.

**Data Availability Statement:** Data derived from public domain resources.

**Conflicts of Interest:** The authors declare no conflict of interest.

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
