# Peer review of "Pricing Product Options and Using Them to Complete Markets for Functions of Two Underlying Asset Prices†"

_jrfm, doi:10.3390/jrfm14080355_

Round 1

Reviewer 1 Report

Thank you very much for giving me the opportunity to review this article ("Product Options").

The structure of the paper, the method used, the data collected and processed reveal an adequate scientific level.

However, I recommend:

The title should be extended in a few words, to be more suggestive to the reader about the content of the paper. The aim is for the reader to quickly understand what he would find in the article, just by reading the title of the article.

Reviewer 2 Report

This manuscript extends the methods to recover the density from derivatives of option prices to two dimensions. The case of product options is studied by employing three models (BSM bivariate normal density, FFT method with multivariate bilateral Gamma, physical measure based on Madan & Wang (2020). The subject is interesting and the obtained formulas can be useful also in other contexts. I recommend acceptance.

Before publishing the article please revise the type-editing in some points. For example:

  • 7, in formulas (41)-(45) the equation labels overlap with the text, and similarly on p.10 and p. 11.
  • Improve the graphical layout of Table 1 and the other Tables introducing vertical spaces and bold characters for the headings.
  • Update the reference item Madan (2020) IJTAF adding volume 23, n. 6, 205004

Reviewer 3 Report

The manuscript deals with product options that are European type derivatives written on two underlying assets. Their payoff at maturity is define as the product of the payoffs of two vanilla options, one on each underlying.
The contributions of the manuscript are essentially theoretical and span the definition of the product, its pricing in the bivariate BS case and its pricing in a general joint characteristic function setup, numerical illustrations and a simulated hedging study are also presented.

Main comments:
- section 2, a sentence stating where the results obtained in this section will be used may help the reader understand the flow of the paper.

- section 3, the development eq. 18 to 25 can be skipped by arguing that in an arbitrage-free market there exists at least on risk-neutral measure and that C(.) can directly be written as eq. (26).

- section 4, the bivariate extension of Breeden-Litzenberger (1978) is similar to the version already obtained for basket options in Lipton (2001) [§8.8.6 p. 291-292] and in Carr & Laurence (2011) [§2.1 eq. (2.8)]. The links between these existing results and the results presented in the manuscript should be mentioned.

Minor comments:
1. page 3, "It may then written"
2. pages 4,7,10,11 ... overlap between formulas and equation numbers should be avoided
3. page 15, in table 1 the values of omega_1 and omega_2 could be given as well
4. page 16, "marlet prices"

References:
- Lipton A. (2001), Mathematical Methods for FX - A financial engineer's approach, World Scientific
- Carr P. & Laurence P. (2011), MULTI-ASSET STOCHASTIC LOCAL VARIANCE CONTRACTS, Mathematical Finance, Vol. 21, No. 1 (January 2011), 21–52
